# *Herpesviridae*, Neurodegenerative Disorders and Autoimmune Diseases: What Is the Relationship between Them?

**DOI:** 10.3390/v16010133

**Published:** 2024-01-17

**Authors:** Maria Antonia De Francesco

**Affiliations:** Department of Molecular and Translational Medicine, Institute of Microbiology, University of Brescia-ASST Spedali Civili, 25123 Brescia, Italy; maria.defrancesco@unibs.it; Tel.: +39-030-399-5860; Fax: +39-030-399-6071

**Keywords:** autoimmune, neurodegenerative, herpesviruses

## Abstract

Alzheimer’s disease and Parkinson’s disease represent the most common forms of cognitive impairment. Multiple sclerosis is a chronic inflammatory disease of the central nervous system responsible for severe disability. An aberrant immune response is the cause of myelin destruction that covers axons in the brain, spinal cord, and optic nerves. Systemic lupus erythematosus is an autoimmune disease characterized by alteration of B cell activation, while Sjögren’s syndrome is a heterogeneous autoimmune disease characterized by altered immune responses. The etiology of all these diseases is very complex, including an interrelationship between genetic factors, principally immune associated genes, and environmental factors such as infectious agents. However, neurodegenerative and autoimmune diseases share proinflammatory signatures and a perturbation of adaptive immunity that might be influenced by herpesviruses. Therefore, they might play a critical role in the disease pathogenesis. The aim of this review was to summarize the principal findings that link herpesviruses to both neurodegenerative and autoimmune diseases; moreover, briefly underlining the potential therapeutic approach of virus vaccination and antivirals.

## 1. Introduction

The *Herpesviridae* family, which includes three subfamilies (α-, β- and γ-herpesviruses), is constituted of enveloped double-stranded DNA viruses distributed worldwide and responsible for various clinical pictures [1]. Herpes simplex virus 1, herpes simplex virus 2 (HSV-1, HSV-2), and varicella-zoster virus (VZV), which belong to the α-virus subfamily, are associated with herpes labialis, genital herpes, chickenpox, and herpes zoster, respectively. Cytomegalovirus (CMV), HHV-6, and HHV-7 are β-herpesviruses, which are associated with congenital/neonatal infections and skin rash (roseola), respectively. Epstein–Barr virus (EBV), the etiological agent of infectious mononucleosis, and Kaposi sarcoma-associated herpesvirus (HHV-8) belong to γ-herpesviruses [1,2]. All of them give rise to lifelong infections in the affected subjects persisting in a latent phase with periodic reactivation. 

Neurodegenerative and autoimmune diseases share a disruption of immune system functions characterized by inflammatory and autoimmune processes. It has also been found that Alzheimer’s disease and multiple sclerosis exhibit a common pro-inflammatory signature [3]. Furthermore, an aberrant activation of inflammasome proteins, important players in innate immunity and inflammation, has been detected in both neurodegenerative and autoimmune diseases such as Alzheimer’s disease, Parkinson’s disease, systemic lupus erythematosus, and multiple sclerosis [4]. Moreover, an important role has been attributed to the B cells in the pathophysiology of both autoimmune and neurodegenerative diseases [5]. The biological characteristic involved in the establishment of herpesvirus latency supports the hypothesis that they might be implicated in the pathogenesis of some neurodegenerative and autoimmune diseases.

They might be the nexus between immune dysfunctions and genetic factors contributing to the pathology of these diseases.

Therefore, this review summarizes the findings supporting a role played by herpesviruses in these diseases and explores all the mechanisms establishing a link between them.

## 2. Neurodegenerative Diseases

Neurodegenerative processes are the most frequent cause of cognitive decline. 

To date, the etiology of Alzheimer’s disease (AD) and Parkinson’s disease (PD), the most common forms of dementia, is not yet well understood. Different risk factors have been associated with both diseases, such as cigarette smoking, caffeine intake, chemical exposure, and diet [6,7]. Furthermore, bacteria, viruses, and lastly prions have been included as risk factors [8,9,10].

AD and PD affect >50 million and >10 million people worldwide, respectively [11,12,13]. Clinically, affected patients manifest behavioral and cognitive alterations with a change in personality [14,15] and autonomic dysfunction [16]. The pathologic mechanism of both diseases involves the accumulation of misfolded proteins in the brain [17,18].

Alzheimer’s disease is characterized by an extracellular deposition of the amyloid b peptide (Aβ) that exists in different forms after the cleavage of the amyloid protein precursor (APP) [19]. This protein plays an important role in central nervous system (CNS) homeostasis because it is involved in synapsis, calcium homeostasis, metal ion capture, and neurogenesis [20]. A high production of Aβ-peptide leads to the formation of neurofibrillary tangles (NFTs) composed of the hyperphosphorylated Tau protein (p-Tau) [21,22]. This results in the destabilization of associated microtubules, synaptic loss, and neurodegeneration [23]. Besides the formation of amyloid plaques and NFTs, AD pathologic mechanisms include oxidative stress and the alteration of calcium homeostasis [24,25], defective autophagy [26], mitochondrial dysfunction [27], and neuroinflammation [28,29]. 

When the disease is acquired before the age of 65, it is considered as early onset AD (EOAD) and it is associated with mutations in genes coding APP, presenilin-1 (PSEN1), and presenilin-2 (PSEN2) [30].

Parkinson’s disease is characterized by a loss of dopaminergic neurons, which are localized mostly in the substantia nigra, but also in the dorsal motor nucleus of the vagus and peripheral neurons [31,32], and by the intraneural deposition of aggregates constituted of Lewy bodies, which are formed by α-synuclein and ubiquitin, and by chronic neuroinflammation [33]. 

An interrelation of genetic, aging, and environmental factors might have a role in their accumulation in neurons [33,34], such as the D620N mutation in a gene and the vacuolar sorting protein 35 (VPS35), involved in PD pathogenesis [35,36,37]. VPS35 belongs to the retromer complex, which acts by forming vesicles that transport cargo molecules from the endosome membrane to the trans-Golgi network (TGN) [38,39]. The D620N mutation was associated with a perturbation of cargo molecule trafficking, a reduction in autophagy, and an accumulation of α-synuclein [40,41,42].

### 2.1. Herpesviruses’ Role in the Pathogenesis of Neurodegenerative Diseases

The postulated hypotheses consist of both a direct and indirect effect on brain. The first assumes a direct viral infection of the brain through the blood and the blood–brain barrier (BBB). In addition to these infection routes, viral entry may occur via monocyte-macrophage/microglia cells [43], the circumventricular organs (CVOs) [44,45], the olfactory bulb, and the peripheral nerves [46].

After entering the central nervous system, viruses start to replicate and activate an inflammatory response that induces microglia cells to produce inflammatory cytokines [47,48] associated with neuronal death also in the substantia nigra (PD) and to neurogenesis [47]. Then, the production of inflammatory cytokines after viral infection leads to the increased expression of the b- and g-secretase enzymes responsible for the cleavage of APP in Aβ-peptide [49] and to the increase in kinases activity related to Tau hyperphosphorylation [50,51].

The indirect effect is based on the viral induction of peripheral inflammation, a role mostly played in AD. The brain interacts with the periphery through the neuronal and humoral pathways [47]. In PD pathogenesis, autoimmune mechanisms induced by molecular mimicry between some herpesviruses and alpha-synuclein are postulated [52] (Figure 1 and Figure 2). 

### 2.2. Research Data That Establish a Link between AD and Herpesviruses

Numerous studies support the involvement of HSV-1 in AD. For example, it was demonstrated that mice infected with HSV-1 showed an accumulation of Aβ-peptide and cognitive impairment [53,54,55].

Moreover, various research papers have shown the presence of HSV-1 DNA in temporal and frontal brain of AD patients compared to matched controls [56,57,58]. In particular, the virus’ genome is principally localized in amyloid plaques [59].

Furthermore, various studies have supported evidence that HSV-1 might induce both direct and indirect inflammation [47,48,49,50,51].

Then, it was shown that HSV-1 induced the production of both Aβ and tau protein in human neural cells [55]; the infection of neuronal and glial cells has been shown to modulate autophagy, which might lead to the accumulation of both Aβ and tau protein [60,61,62,63]. HSV-1 infection seems to influence the autophagic process through the binding of the HSV1 ICP34.5 (infected cell protein 34.5) to Beclin1, a protein involved in the synthesis and maturation of autophagosomes [64]. Therefore, this interaction blocks the host autophagy response [65]. Moreover, the binding of the HSV-1 US11 protein to the protein kinase PKR prevents its activity [66]. Next, it has been shown that HSV-1 infection increases the production of reactive oxygen species (ROS) [67], which in turn enhances the impairment of the autophagic process determined by the HSV-1 infection [68] creating a vicious circle.

Patients carrying the apolipoprotein E (*APOE*) E4 allele, a genetic risk factor for AD, had a greater risk of developing AD [69,70]; in addition, HSV-1 DNA has more often been detected in the brains of these subjects [71]. Serological studies have also detected the presence of Aβ and HSV-1 cross-reactive antibodies in patients with AD [72,73], and, furthermore, two prospective studies involving 512 and >8000 subjects, respectively, showed that increased levels of IgM antibodies against HSV-1,2 and symptomatic infection with these viruses were associated with a hazard ratio (HR) of 2.55 and a three-fold increased relative risk, respectively, of developing dementia later in life [74,75]. Furthermore, different studies have supported an important role of EBV and HHV-6 in AD pathogenesis. In fact, EBV DNA has been found in the blood of 45% of patients with AD; then, the virus genome has been found in the brain of 6% of AD patients carrying the pathogenic apolipoprotein E (*APOE*) E4 allele [76].

Furthermore, EBV induces an inflammatory environment as demonstrated by the evidence that EBV reactivation from the latency phase gives rise to a systemic immune response, with an increase in the inflammation processes associated with cognitive decline in older age [77]; further, that B lymphocytes derived from a patient with AD and infected with EBV produce high levels of TNF-α in vitro, leading to accumulation of Aβ protein and to the hyperphosphorylation of the tau protein [78,79]. Subsequently, CD8 effector memory CD45RA+ (TEMRA) cells have been found to be responsible for proinflammatory and cytotoxic functions in AD patients and to play an important role in disease progression; two antigens of EBV (the Epstein–Barr nuclear antigen 3 and the trans-activator protein BZLF1) have been identified as stimulators of this adaptive immunity in AD patients [80,81].

The EBV protein, BNLF-2a, has been found to be involved in AD progression because it blocks the transporter protein (TAP) associated with antigen processing and decreases the expression of the major histocompatibility complexes (MHC)-I and II, which might determine the increase in neuronal cells and viral polypeptides in the environment [82,83].

Additionally, HHV-6 might promote neuroinflammation. In fact, HHV6-infected primary glial cell cultures produced high levels of proinflammatory cytokines [84]; furthermore, microglial and T cells infected with HHV6 showed an increased expression of Aβ 1-42, tau and ApoE proteins [85,86].

Furthermore, autophagy was reduced following the infection of astrocytoma cells and primary neurons with HHV-6 and this induced an increase in the misfolding of the Aβ protein and hyperphosphorylation of the tau protein [87,88,89].

A study comparing the transcriptome of white blood cells (WBCs) infected with HHV6 to the transcriptome of WBCs derived from AD patients showed the sharing of 95 differentially expressed genes, mostly involved in antigen presentation by MHC class II antigens [90]. The same study showed that the host response against CMV, EBV, and HHV6 involved oxidative stress mechanisms activating sirtuin-1 and the peroxisome proliferator-activated receptor-g coactivator (PGC)-1 pathway [90].

However, the first evidence that HHV6 DNA was present at higher levels in AD brains than in healthy controls [91] was confuted by other successive studies employing the same RNA data set [92,93,94].

On the other hand, few epidemiological and serological studies reported an association between AD and VZV and CMV.

Epidemiological studies involving 80,000 and 3384 patients with herpes zoster showed a significantly increased risk (HR_1.11; 95% CI, 1.04 to 1.17 and HR_2.83; 95% CI, 1.83 to 4.37, respectively) of developing dementia during aging [95,96]; therefore, VZV might contribute to AD pathogenesis through its binding to the insulin-degrading enzyme (IDE), a zinc metalloprotease associated with Aβ degradation, inducing an impairment of its activity [97,98].

Epidemiological studies showed an association between CMV and moderate–severe dementia [99]. Longitudinal studies showed that higher percentages of patients with increased CMV blood markers exhibited an impairment of cognitive functions after a 4–5-year period [76,100]. Finally, it was shown that an increased density of neurofibrillary tangles is associated with the plasmatic levels of CMV IgG antibodies [101].

### 2.3. Research Data That Establish a Link between PD and Herpesviruses

Different immunological and virological studies correlate HSV1, EBV, HHV-6, and CMV to PD.

A molecular mimicry between HSV-1 and α-synuclein produces autoimmune responses which trigger aggregation of α-synuclein, target neurons of substantia nigra, and induce subsequent neuronal degeneration [102,103]; therefore, the production of TNF-α following HSV-1 infection might contribute to PD pathogenesis. In fact, dopaminergic neurons are very susceptible to TNF-α activity, which induces neuronal death [103].

Molecular mimicry between the C-terminal region of α-synuclein and a repeat region in latent membrane protein 1 (LMP1) encoded by EBV produces oligomerization of α-synuclein [104].

CMV reactivations could expedite the onset of PD inducing a neuroinflammatory environment with the production of inflammatory cytokines by dendritic cells [105]; furthermore, these cells might present antigens derived from dopaminergic neurons, which might be responsible for autoimmune response to neuromelanin [106].

HHV-6 might contribute to PD pathogenesis by direct CNS entry, immunologically mediated mechanisms or inducing parainfectious cytotoxic changes [107]. Finally, a study that analyzed the transcriptome datasets from seropositive or seronegative patients for CMV, EBV, and HHV6 and PD patients showed that patients infected with these herpesviruses shared the differentially expressed genes *BCL6*, *GYG1*, *RBCK1*, *TIMP2*, and *CIRBP* with PD patients [90]. In particular, TIMPs (tissue inhibitors of metalloproteinases) are inhibitors of matrix metalloproteinases (MMPs), the altered expression of which is related to neuroinflammation and neuronal death [108].

The principal findings supporting a role for different herpesviruses in the etiology of the described neurodegenerative diseases are summarized in Table 1. 

## 3. Autoimmune Diseases 

At present, 80 autoimmune diseases are known to affect about 5% of the general population [109]; because most of them have unknown etiology, different factors such as genetics, environment, age, and viruses are considered to be triggers of aberrant immune responses [110,111]. Among the many known autoimmune diseases, only three (multiple sclerosis, systemic lupus erythematosus, and Sjögren’s syndrome) have been included in this review because of the association found between them and herpesviruses.

Multiple sclerosis (MS) is a chronic inflammatory disease of the central nervous system responsible for severe disability in young adults [112]. The estimated prevalence is 2.8 million people worldwide [113]. A dysregulated immune response is at the root of myelin destruction that covers axons in the brain, spinal cord, and optic nerves leading to demyelination and axonal degeneration [112]. The disease starts with a relapsing–remitting clinical form and becomes progressively chronic in later clinical phases [114]. The etiology of MS is heterogeneous including an interrelationship between genetic factors, principally immune associated genes, and environmental factors such as infectious agents, vitamin D deficiency, sun exposure, obesity, and smoking [115,116,117].

Systemic lupus erythematosus (SLE) is a composite autoimmune disease [118] characterized by the alteration of B cell activation leading to the production of multiple autoantibodies, dysregulation of T cell function with impairment of cell-mediated immunity [119], impaired clearance of nucleic acids, and increased Type 1 IFN response [120]. Characteristic serologic markers of SLE are antinuclear antigens (ANA), anti-double-stranded DNA (dsDNA) and anti-Smith (anti-Sm) autoantibodies [120]. Clinical symptoms range from initial musculoskeletal and mucocutaneous symptoms to the later involvement of any system [121,122]. Genetic and environmental factors such as infectious agents have been recognized to play an important role in its pathogenesis [123,124,125].

Sjögren’s syndrome (SS) is a heterogeneous autoimmune disease [126] characterized by inflammatory infiltration and autoimmune response against exocrine glands [127,128], aberrant polyclonal B cell activation with the production of autoimmune antibodies against ribonucleoproteins [129,130]. Its prevalence is estimated between 0.2 and 0.5% and it mostly affects middle-aged women [131]. At onset, clinical manifestations of primary Sjögren’s syndrome include chronic fatigue and mucosal dryness of the mouth and eyes. The progression of primary Sjögren’s syndrome to systemic disease involves different organs leading to interstitial lung disease, autoimmune cholangitis, hepatitis, vasculitis, and alteration of both the peripheral and central nervous system [132].

Like in other autoimmune diseases, in Sjögren’s syndrome different factors involving genetics, immune responses, and the environment play important roles in disease development [132]. 

### 3.1. Herpesviruses’ Role in the Pathogenesis of Autoimmune Diseases

Different mechanisms are postulated as possible effects of viruses in generating a disruption of immune system response, but the principals involve molecular mimicry, bystander activation, and epitope spreading.

Molecular mimicry implies a structural, functional, or genetic similarity between viral proteins and host proteins. This similarity between virus and host can generate immune responses by activating autoreactive T and B cells capable of destroying both self and non-self-antigens [133,134,135].

Bystander activation consists of a production of a pro-inflammatory environment with activation of dendritic cells and autoreactive naïve T cells inducing damage to healthy cells and the release of self-antigens able to activate autoimmune reactions [136].

Epitope spreading relies on the release of self-antigens during the course of a viral infection and on the activation of autoreactive cells, which target self-epitopes [137] (Figure 3 and Figure 4).

### 3.2. Research Data That Establish a Link between Multiple Sclerosis and Herpesviruses

Several studies have shown a direct and indirect role of HSV-1 in inducing demyelination.

Some studies reported that HSV-1 DNA has been more frequently detected in the peripheral blood and in the brain of MS patients than in the brain of controls [138,139].

Moreover, HSV-1 infection of mice induced both CNS demyelination and inflammation [140]. Interestingly, it was found that these effects were related to the murine strains. Similarly, it was found that children that lack the DRB1*15 allele and had HSV-1 seropositivity were associated with an increased risk of MS [141], underlining that genetic background is important for the development of MS [142].

Furthermore, it has been shown that HSV-1 might induce an increase in brain–blood barrier (BBB) permeability during both acute infection and latency. Microglia infected by HSV-1 release inflammatory cytokines such as TNF-a, IL-1b and IFN-g, able to increase the expression of ICAM1 and NO levels altering endothelial cell function [143]. In addition, chronic immune responses in infected neuronal cells are present during HSV-1 latency in the trigeminal ganglia [144] and they might generate a persistent inflammatory environment leading to the production of autoreactive T cells [145]. Finally, it was found that HSV-1 infection generated mitochondrial dysfunction and the following release of reactive oxygen species (ROS) is associated with neurodegenerative processes [146,147,148]. Therefore, the interference of HSV-1 with the autophagy in the CNS could lead to the accumulation of cellular proteins and of myelin debris [149].

Then, another way by which the virus could have a role in MS pathogenesis is via molecular mimicry. In fact, autoreactive T cells might induce CNS inflammation by binding epitopes in the brain, which share molecular mimicry with viral antigens. To support this hypothesis, different studies have shown the presence of T cells and antibodies able to cross-react with the HSV-1 protein UL15, with the antigen myelin basic protein, and with the HSV-1 glycoprotein B and a brain epitope, respectively [150,151].

Among the herpesviruses, EBV is one that has been investigated the most for its role in triggering MS. Different mechanisms have been considered, such as molecular mimicry, defective EBV immune responses, and EBV-induced inflammation.

#### 3.2.1. Molecular Mimicry

Different EBV antigens share epitopes similar to host proteins, leading to both cross-reactive humoral and cellular immune responses. Different studies have shown that T cells autoreactive to basic myelin proteins derived from MS patients were cross-reactive to several viral peptides such as EBNA-1 [152,153]. Furthermore, structurally related pairs of peptides from EBNA-1 and b-synuclein, a protein present in the brain and involved in MS, have been identified [154]. Then, molecular mimicry between anoctamin 2 (ANO2), a chloride channel in the brain, and EBNA-1 has been associated with an increased risk of MS [155]. Other cross-reactivities have been found between EBNA-1 and α-cristallin B chain (CRYAB) [156] and the glial cell adhesion molecule [157]. Additionally, molecular mimicry has been found between EBV lytic proteins BHRF1 and BPLF1 and the self-peptide derived from the RAS guanyl releasing protein 2 (RASGRP2) [153], expressed both in B cells and in neurons, and responsible for generating autoreactive T cells. These autoreactive CD4+ T cells enter the brain, where they induce an inflammatory response leading to demyelination and axonal damage [158].

#### 3.2.2. Defective EBV Immune Responses

Generally, cytotoxic CD8+T cells control EBV-infected B cells, but this mechanism is defective in MS patients [159]. So, virus-infected B cells become resistant to apoptosis [154] and generate co-stimulator signals that activate autoreactive T cells. Then, B cells can contribute to disease progression by acting directly in the CNS. Even if perivascular MS brain lesions show a low number of B cells, their presence is high in the MS meningeal site and is associated with cortical damage [160]. Antibodies produced by B cells, that have been boosted in the CNS, are oligoclonal IgG against EBV [161]. Higher titers of EBNA-1 antibodies have been found in both the serum and CSF of MS patients and they have been associated with a higher risk of MS [162,163], a risk that is more elevated in subjects with HLA class II DR2b (DRB1*1501 b, DRA1*0101 a) [164]. Furthermore, a prospective study showed a four-fold increase in anti-EBNA 2 antibody titers in MS patients [165]. In large epidemiological studies, EBV seroconversion has been found to precede the outcome of MS clinical signs [166] and another large prospective study found that all of the MS patients included in the analysis were positive for EBV infection [167].

All of these serological studies support evidence for dysregulated EBV immune responses in MS.

#### 3.2.3. EBV Induced Inflammation

EBV remains in a latent state in memory B cells, which exhibit a pro-inflammatory phenotype. Therefore, they produce various inflammatory cytokines both in meninges and in regional lymph nodes, in addition to an inflammation process mediated by the release of exosomes containing EBV-encoded immunomodulatory RNAs (EBER1, miRNA) [168]. These exosomes stimulate cellular functions such as dendritic cells’ antiviral inflammatory activity [168], while miRNAs affect genes associated with MS risk [169]. Additionally, the subsequently created inflammatory environment triggers a BBB permeability that allows anti-EBV immune cells across, the activation of both microglia and astrocytes, and a possible oligodendrocyte dysfunction, which together contribute to neuronal destruction [170,171,172,173,174].

Different studies have demonstrated an association between HHV-6 and MS. Indeed, higher levels of HHV-6 mRNA and viral proteins have been detected in MS plaques rather than in the white matter in the brain of patients compared to controls [175], in particular in oligodendrocytes. Higher levels of the HHV-6 genome have been found mostly in acute rather than chronic lesions in MS patients who did not receive immunomodulatory therapies [176]. A strong association with an OR of 6.7 and a 95% CI of 4.8 to 8.6 (*p* < 0.00001) [177] has been found between HHV-6 and MS in a meta-analysis of 39 studies. Furthermore, an association has been found between HHV6 reactivation and disease activity in relapsing–remitting MS (RRMS) and secondary progressive MS (SPMS) [178]. Then, different studies have shown that serological response against HHV-6 is higher in MS patients than in controls [179,180]. In addition, a Swedish study analyzing 8742 MS subjects and 7215 matched controls showed an increase in IgG response against the IE1A peptide of HHV-6A [181]. Then, an increase in HHV-6-specific antibodies was associated with clinical relapses [182], while raised anti-HHV-6 early antigen (p41/38) IgM antibodies have been shown in patients with RRMS as compared to patients affected by other MS subtypes or other neurological/autoimmune diseases [183]. Oligoclonal bands against HHV-6 have been detected in the CSF of MS patients underlining the production of HHV-6-specific IgG [184]; furthermore, intrathecal HHV-6 antibodies have been more frequently identified in RRMS and chronic progressive MS than in other neurological diseases [185].

Additionally, a sequence similarity has been identified between the U24 protein of HHV-6 (residues 4–10) and MBP (residues 96–102) [186]. Furthermore, the frequency of circulating T cells recognizing both HHV-6 U24 and MBP is higher in MS patients than in controls [186,187]. Furthermore, HHV-6-infected T cells induce the increased production of different inflammatory cytokines, which in turn correlates with the severity and progression of MS [188]. In addition, HHV-6 infection of oligodendrocyte precursor cells affects the remyelination process in MS patients [189], impairing the differentiation and migration process of these cells [190]. Furthermore, for HHV-6, a role as trans-activator of latent viruses such as EBV or endogenous retroviruses [191,192], which also play an important role in MS pathogenesis, has been proposed. Lastly, increased levels of a soluble CD46 receptor, a complement system regulator, and the human receptor for HHV-6, have been found in the CSF and blood of MS patients [193], leading to the hypothesis that HHV-6 infection, through the engagement of CD46, might induce exaggerated activation of the complement system which could contribute to MS pathogenesis [194]. Another supposition was that during its replication, HHV-6 might incorporate different host antigens, therefore provoking autoimmune responses [194].

The role of HCMV in MS is quite controversial, because several studies proposed both a deleterious and protective HCMV role.

In fact, some studies have shown higher loads of HCMV genome in MS patients than healthy controls [195,196]; furthermore, HCMV has been found in the plaques and the CSF of MS patients [197].

Then, it has been shown that reactivation of HCMV was associated with a deterioration of the course of MS in some subjects [198]. 

Moreover, a similarity has been shown between the HCMV antigen (UL86981-1003) and myelin oligodendrocyte glycoprotein (MOG) (residues 34–56), supporting the hypothesis about the generation of autoreactive T cells [199].

By contrast, other studies support a beneficial role of HCMV in MS. In fact, a recent study has shown a lower prevalence of IgM against HCMV in MS patients than in controls. Furthermore, the patients had relapsing MS, underlining that HCMV has a part in reducing disease severity [200].

Furthermore, it has been shown that HCMV infection is related to a decreased production of inflammatory cytokines in progressive MS, confirming a protective role of HCMV in MS [201].

Few epidemiological and serological studies shown an association between VZV and MS.

A study conducted in Taiwan showed that patients with herpes zoster had a risk for MS of 3.63-fold higher than controls [202]; furthermore, MS patients showed specific antibodies against VZV [203]. In addition, a meta-analysis involving 2266 MS patients and 1818 healthy subjects showed that VZV seropositivity was higher in MS patients than in controls (OR = 4.47, *p* < 0.001) [204]. Moreover, high loads of VZV DNA have been detected in the CSF and PBMCs of MS patients [205]. Finally, a recent Mendelian randomization analysis was used to study the association between VZV and MS by using summary statistics from genome-wide association studies (GWAS). The results supported a significant association between genetically predicted chickenpox and the risk of MS with an OR of 35.27 (CI = 22.97–54.17, *p* = 1.46 × 10^−59^) [206].

### 3.3. Research Data That Establish a Link between Systemic Lupus Erythematosus and Herpesviruses

Serological studies have found a strong association between SLE and EBV and, to a lesser extent, between SLE and CMV. In particular, SLE patients presented an increased reactivation of EBV, as supported by the detection in their blood of high loads of EBV DNA [207] and a significantly higher OR for the presence of IgG against VCA and EA in SLE patients than in controls (OR = 2.06, 95% CI= 1.30–3.26, *p* = 0.002 and OR = 7.70, 95% CI= 4.64–12.76, *p* < 0.001, respectively) [208].

Different studies have detected higher anti-EBV antibody titers in SLE patients compared to healthy subjects [209,210,211]; however, a significant correlation between EBV serology and single nucleotide polymorphisms (SNPs) in genes related to SLE has been found [212], underlining the important role played by genetic factors in SLE pathogenesis. In addition, different cross-reactivities have been detected between EBV antigens and SLE autoantigens, leading to the production of autoantibodies such as anti-EBNA 1 antibodies that cross-react with autoantigens of SLE (SmB, SmD and Ro) [213]. Then, anti-EBNA1 antibodies have been found to cross-react with dsDNA in mice [214] and with the C1q complement component [211]. In fact, SLE patients who had a seropositivity against EBNA348, a peptide of EBNA1, showed higher titers of anti-C1q [215]. A cross-reactivity has been further found between EBNA2 and the antigenic terminal domain of the SLE antigen SmD1 [216]. In addition, it has been shown that EP4, a peptide from EBV EA, induced an increase in anti-SmD and anti-Ro and correlated with the SLE disease activity index (SLEDAI) [217]. Finally, EBV IL10, a viral EBV gene expressed during its lytic cycle, is homologous of IL10 and, because it engages the same receptor, it inhibits all the immune responses regulated by IL10 [218]. 

vIL-10 has been detected at higher concentrations in the plasma of SLE patients than in controls [219] and it has been correlated to an increased production of inflammatory cytokine leading to a defective clearance of infected cells and to increased antigen presentation that may generate autoimmune responses [218,220].

On the other hand, for CMV, a similarity has been detected between an epitope of phosphoprotein 65 and TATA-box-binding protein associated factor 9 (TAF9134-144) leading to the production of antibodies cross-reacting with both proteins, ANA and anti-ds DNA in mice [221]. Then, an increase in antibody titers against TAF9 has been shown in SLE patients [222]. Furthermore, a monoclonal antibody against the CMV UL44 immunoprecipitated both viral UL44 and some SLE antigens such as nucleoli, dsDNA, and ku70 [223]; and, SLE patients have displayed an increased level of CMV US31 leading to macrophage differentiation and inflammation activation [224].

A cross-reactivity between CMV-specific T cells and the La protein has been detected at the beginning of SLE in childhood [225]. Moreover, CMV antigens may induce, in the PBMCs of SLE patients, an increase in IFN-gamma and IL-4 cytokines with a greater expansion of memory T cells potentially involved in autoimmune events [226].

There is no evidence that HHV-6 has a role in the occurrence of SLE. The only hypothesized role indicated for this herpesvirus was the establishment of a coinfection with EBV determining a synergistic effect that might generate an activation of polyclonal B lymphocytes and a perturbation of immune functions [208].

### 3.4. Research Data That Establish a Link between Sjögren’s Syndrome and Herpesviruses

Different studies have shown an association between SS and EBV, VZV, and HHV-6.

In fact, some studies have shown a correlation between EBV reactivation in SS patients and B cell polyclonal activation, which contributes to autoantibody production [227,228]. Then, it has been shown that EBV infection plays a role in B and T cell differentiation in an effector phenotype related to SS. Lymphocyte activation at ectopic germinal center, induced by follicular T helper and cytotoxic cells, and probably stimulated by EBV, might give rise to autoimmune epithelitis [229]. High levels of EBV DNA have been found in the salivary gland tissue and PBMCs of SS patients [230]; further, high levels of anti-EA have been found in SS patients [231]. By analyzing the gene expression of PBMCs derived from both SS patients and healthy controls, it has been shown that several differentially expressed genes were shared, including abnormal signaling pathways of T and B cell receptors and virus-correlated pathways [232]. Finally, similarities between viral EBNA 2 and the Ro-60 antigen and between EBER-1 and the La antigen have been detected [233].

A recent nation-wide, population-based, case–control study including 5751 SS patients and 28,755 matched controls showed a significant correlation between herpes zoster exposure and SS risk, which was greater when the interval between the last visit for herpes zoster infection and the index date was <3 months (OR adjusted for comorbidities = 3.13, 95%CI = 2.20–4.45) [234].

Higher titers of anti-HHV-6 have been found in a group of SS patients with rheumatoid arthritis than in controls [235]; however, even if HHV-6 might contribute to SS development, the contrary is also true. Namely, the autoimmune disease might lead to the reactivation of HHV-6 [236].

To date, it is not known if CMV might contribute to SS development. The virus reactivation observed in SS patients might be due to the use of pronounced immunosuppression [237].

The principal findings supporting a role for different herpesviruses in the etiology of the described autoimmune diseases are summarized in Table 2. 

## 4. Relationship between Herpesviruses, Neurological Disorders and Autoimmune Diseases

From the high number of reported studies, how can we relate the same herpesviruses to pathologies as distinct as neurodegenerative and autoimmune diseases? All of them share the biological characteristic to persist in the host in a latency state interspaced with periodic reactivation. 

This periodic reactivation might lead to chronic pathological conditions. Is the virus’ reactivation a consequence of the immune dysregulation linked to these diseases or is the virus infection a trigger of neurodegenerative and autoimmune diseases? The starting point is quite controversial and probably only longitudinal studies might clarify this issue. However, a recent study found 45 significant associations between the exposure to a viral infection and the risk of contracting a neurodegenerative disease later in life [238]. Among the viral infections, viral encephalitis and varicella zoster are included, supporting the results of previous studies, which found associations between HSV encephalitis and AD [239,240], HSV and MS [241], EBV and dementia [242], and EBV and MS [166].

Is there a common pathogenesis mechanism involved in both neurodegenerative and autoimmune diseases induced by herpesviruses?

Molecular mimicry was one of the mechanisms postulated for PD pathogenesis involving HSV1 and EBV [102,103,104]; the same mechanism was invoked for MS involving HSV1 [150,151], EBV [154,155,156,157,158] and HCMV [199], and also for SS involving EBV [233]. Molecular mimicry was finally detected between EBV and CMV proteins and self-antigens in SLE [213,214,215,221,225].

Another mechanism underlying pathogenesis in neurodegenerative and autoimmune diseases is herpesvirus-induced peripheral and central inflammation. This was reported for HSV-1 [47,48,49,50,51], EBV [77,78,79], and HHV-6 [84] in AD; for HSV1 [103], CMV [105], and HHV6 [106] in PD; for HSV1 [140,143], and EBV [168,169,170,171,172,173,174] in MS, and for EBV [218,220] and CMV [226] in SLE.

What is the important trigger that directs the development of a neurodegenerative or an autoimmune disease from the same herpesvirus?

It is likely that host-specific factors play a pivotal role in the regulation of virus virulence. Given the ubiquity of herpesviruses compared to the lower frequency of these diseases, it is obvious that not all herpesvirus-infected subjects develop neurodegenerative or autoimmune diseases. There may be both virus- and host-related factors that have an influence on the host immune response and/or on the susceptibility to virus-induced pathological effects. A recognized important host-related factor is genetic background, supported by evidence of specific host genetic loci that increase the risk of the development of neurodegenerative or autoimmune diseases, such as the presence of APOE4, related to increased AD susceptibility [243]. Then, a recent study investigating the shared genetic susceptibility between AD and MS detected 16 shared loci, with 8 of them able to have concordant effects both on AD and MS [244]. These genes were involved in molecular signaling pathways related to inflammation and the structural organization of neurons. Therefore, such genetic factors could contribute in a determining way to what fate will befall infected subjects, in terms of developing one or the other pathologic condition.

## 5. Conclusions 

The etiology of both neurodegenerative and autoimmune diseases is complex and heterogenous. 

However, even if it is not possible to discriminate whether herpesviruses are the cause or the effect of these diseases, the overlapping of pathogenetic pathways between virus infections and neurodegenerative and autoimmune processes constitutes a motivation to use antiviral therapies as treatment approach for these diseases. To date, in spite of promising results [244,245], few data are available on the efficacy of antiviral therapies.

Then, vaccination has also been proposed to mitigate the possible risk of developing neurodegenerative diseases by using available or future vaccines against herpesviruses. An efficacious viral vaccine might reduce virus diffusion and limit altered immune responses, that have in turn a consequent role on disease pathogenesis because both inflammation and dysregulated immune responses are known to contribute greatly to these pathologies. Some results support this hypothesis. In fact, varicella-zoster vaccination has been associated with a reduced risk of dementia, AD, and PD in both the United States and Wales [246,247,248,249,250].

Future research is needed to fully understand the correlation between herpesviruses and host-specific factors and their impact on the disease development, and might lead to the identification of novel targets to prevent or slow the progression of both neurodegenerative and autoimmune diseases. 

## Figures and Tables

**Figure 1 viruses-16-00133-f001:**
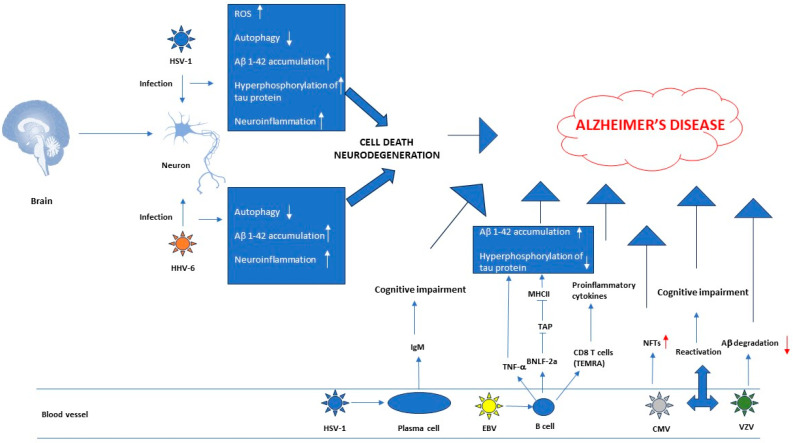
Role of herpesviruses in Alzheimer’s disease. Where there is no explanation next to it, the arrow with the tip indicates activation, the arrow without the tip indicates inhibition. The red up arrow indicates increase, while the red down arrow indicates decrease. Abbreviations: ROS, reactive oxygen species; TAP, transporter protein; NFTs, neurofibrillar tangles.

**Figure 2 viruses-16-00133-f002:**
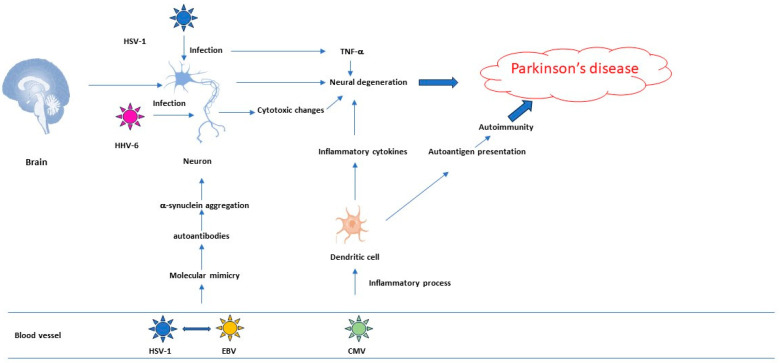
Role of herpesviruses in Parkinson’s disease.

**Figure 3 viruses-16-00133-f003:**
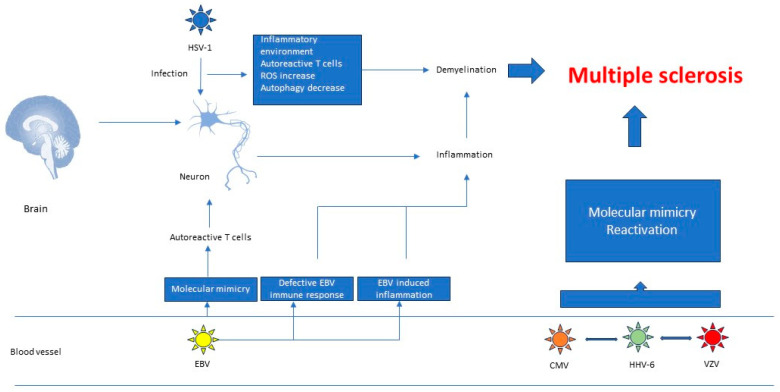
Role of herpesviruses in multiple sclerosis.

**Figure 4 viruses-16-00133-f004:**
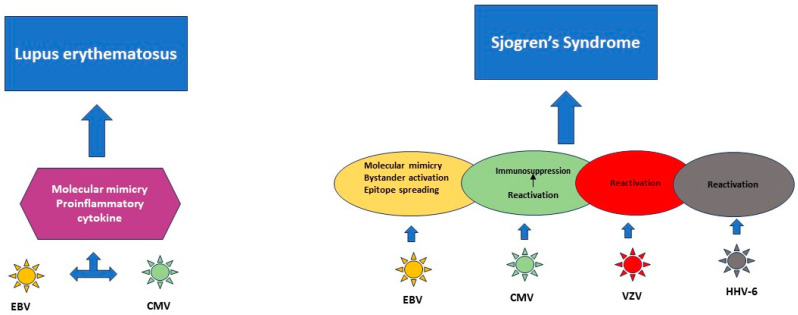
Role of herpesviruses in systemic lupus erythematosus and Sjogren’s syndrome.

**Table 1 viruses-16-00133-t001:** Summary of findings supporting a role of herpesviruses in neurodegenerative diseases.

Disease	Result	Study Type	Refs
AD	The presence of Aβ and HSV-1 cross-reactive antibodies in patients with AD	Comparative study	[73]
	An increased level of IgM antibodies against HSV-1,2 was associated with an increased risk of developing dementia	Prospective cohort studies	[74,75]
	AD-derived B lymphocytes infected with EBV produced high levels of TNF-α in vitro	Cell model studies	[78,79]
	EBV antigens stimulated cytotoxic and proinflammatory functions by CD45RA+ cells	Prospective cohort and knockout studies	[80,81]
	Increased CMV blood markers associated with cognitive decline	Longitudinal, prospective cohort studies	[76,100]
	Levels of CMV IgG antibodies associated with increased neurofibrillary tangles	Longitudinal study	[101]
	Patients with herpes zoster showed a significantly increased risk of developing AD	Retrospective cohort studies	[95,96]
	Studies showed an association between CMV and moderate–severe dementia	Population cohort study	[99]
	HSV-1 induced neuroinflammation	In vitro model studies	[47,48,49,50,51]
	HSV-1 induced the production of both Aβ and the tau protein in human neural cells	In vitro infection studies	[55]
	HSV-1 DNA was present in the brain of AD patients	Molecular studies	[57,58]
	EBV was present in the blood and brains of AD patients	Comparative and molecular study	[76]
	EBV reactivation has been associated with cognitive decline	Longitudinal study	[77]
	EBV BNLF-2a has been associated with AD progression	Biochemical studies	[82,83]
	HHV-6-infected microglia showed an accumulation of Aβ and tau proteins	In vivo and in vitro infection model	[85,86]
	HHV-6 reduced autophagy	In vitro infection model	[87,88,89]
	HHV-6-infected patients and AD patients shared 95 differentially expressed genes	Computational analysis study	[90]
	Binding of VZV to the insulin-degrading enzyme	In vitro infection model	[97,98]
PD	Molecular mimicry between HSV-1 and α-synuclein	Seroprevalence study	[102]
	Molecular mimicry between EBV LMP1 and α-synuclein	Seroprevalence study	[104]
	CMV-, EBV-, and HHV-6-infected patients shared several differentially expressed genes with AD patients	Computational analysis study	[90]

AD, Alzheimer’s disease; PD, Parkinson’s disease, LMP-1, latent membrane protein-1.

**Table 2 viruses-16-00133-t002:** Summary of findings supporting a role for herpesviruses in autoimmune diseases.

Disease	Result	Study Type	Refs
MS	Molecular mimicry between HSV-1 protein and myelin basic protein	Biochemical study	[150]
	Molecular mimicry between EBV protein LMP-1 and different proteins involved in MS pathogenesis	Case–control studies	[152,153,154,155]
	Molecular mimicry between EBV proteins BHRF1 and BPLF1 with a protein present in neurons	Epitope discover approach and cell immunity analysis	[158]
	Defective cytotoxic T cells control of EBV in MS	Seroprevalence study	[159]
	Oligoclonal IgG against EBV have been detected in the brain of MS patients	Comparative study	[161]
	Higher titers of EBV antibodies have been detected in MS patients	Case control study	[164]
	Serological response against HHV-6 was higher in MS patients	Molecular, case–control studies	[179,180,181,182,183]
	Oligoclonal IgG against HHV-6 have been detected in the brains of MS patients	Seroprevalence study	[184]
	Molecular mimicry between HHV-6 protein U24 and myelin basic protein	Seroprevalence study	[186]
	Increased level of sCD46, the receptor for HHV-6 has been detected in MS patients	Immunological study	[193]
	Molecular mimicry between HCMV antigen (UL86981-1003) and myelin oligodendrocyte glycoprotein (MOG)	In vivo experimental model	[199]
	Patients with herpes zoster have been associated with a higher risk of MS	Population-based study and computational GWAS	[202,206]
	HSV-1 DNA in the blood and in the brains of MS patients	Molecular studies	[138,139]
	HSV-1 infection-induced CNS demyelination and neuroinflammation	In vitro and in vivo infection studies	[140,141,142,143,145]
	EBV-induced an inflammatory environment	In vitro infection and molecular studies	[168,169,170,171,172,173]
	High load of HCMV genome has been detected in MS patients	Case–control study	[195]
	High load of VZV genome has been detected in MS patients	Molecular study	[205]
SLE	Serological response against EBV was higher in SLE patients	Case–control studies	[209,210,211]
	EBV antibodies cross-reacted with autoantigens of SLE	Seroprevalence studies	[215,216,217]
	Similarity between CMV antigens and autoantigens of SLE	Animal model and in vitro infection studies	[222,223]
	High load of EBV DNA was detected in SLE patients	Molecular study	[207]
	vIL-10, a viral EBV protein, has been detected at higher concentrations in plasma of SLE patients	In vitro infection model	[219]
SS	Higher titers of anti-HHV-6 have been found in a group of SS patients	Seroprevalence study	[235]
	A significant correlation between herpes zoster exposure and SS risk has been detected	Population-based case–control study	[234]
	High levels of EBV DNA have been detected in SS patients	Molecular study	[230]
	Increased EBV reactivation has been observed in SS patients	Observational study	[228]

MS, multiple sclerosis; SLE, systemic lupus erythematosus; SS, Sjögren’s syndrome; LMP-1, latent membrane protein-1; anti-EA, anti-early antigen.

## Data Availability

Not applicable.

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
