# Peer review of "Herpesviridae, Neurodegenerative Disorders and Autoimmune Diseases: What Is the Relationship between Them?"

_viruses, 2024, doi:10.3390/v16010133_

Round 1

Reviewer 1 Report

Comments and Suggestions for Authors

This manuscript reviews the potential roles of various human herpesviruses in the etiology and pathogenesis of neurodegenerative diseases (Alzheimers, Parkinsons) and autoimmune illnesses (including multiple sclerosis and lupus).  Unfortunately, rather than concentrating on the most compelling and important previously published papers that provide evidence that one or more  herpesviruses might contribute to the above illnesses (for example, the roles that EBV may play in multiple sclerosis), the manuscript instead  lists superficial descriptions of a huge number of published papers.  The author does not focus the review on the virus(es) that are most likely to contribute to these illnesses but instead discusses the potential roles of each of the various different human herpesviruses, implying they are equally likely to contribute to these illnesses. Based upon review of the author's publications listed in Pubmed, it is not clear that the author has  specific expertise in the field of human herpesviruses.  

Comments on the Quality of English Language

This manuscript requires moderate editing to correct gramatical errors and misuse of some words

Author Response

This manuscript reviews the potential roles of various human herpesviruses in the etiology and pathogenesis of neurodegenerative diseases (Alzheimers, Parkinsons) and autoimmune illnesses (including multiple sclerosis and lupus).  Unfortunately, rather than concentrating on the most compelling and important previously published papers that provide evidence that one or more  herpesviruses might contribute to the above illnesses (for example, the roles that EBV may play in multiple sclerosis), the manuscript instead  lists superficial descriptions of a huge number of published papers.  The author does not focus the review on the virus(es) that are most likely to contribute to these illnesses but instead discusses the potential roles of each of the various different human herpesviruses, implying they are equally likely to contribute to these illnesses. Based upon review of the author's publications listed in Pubmed, it is not clear that the author has  specific expertise in the field of human herpesviruses.  

I agree with the reviewer’s comment about the different role played by the various herpesviruses in the considered diseases. I have therefore added a brief sentence for each of them underlining the major or minor role they have in the pathogenesis of neurodegenerative and autoimmune diseases. Because the focus of the review was to consider all the herpesviruses potentially involved in these diseases, a summary of all the described mechanisms for each of them was a choice dictated by the need not to be excessively long-winded.

Reviewer 2 Report

Comments and Suggestions for Authors

In this "review" article, Dr. De Francesco disscussed the possible association between herpes virus with neurodegenerative and autoimmune diseases. I do have some questions and concerns:

1) The author did not explain why neurodegenerative and autoimmune were picked? Are they interconnected? Please explain in the text and abstract.

2) Similarly, why among many autoimmune diseases, SLE, MS and Sjogren were selected?

3) Please rephrase the subheadings for example "How might herpesviruses contribute to the pathogenesis of these neurodegenerative diseases?" to make it more suitable for a conventional review article. Otherwise, this should be a perspective article, not a review.

4) Please add tables (one for neurodegenerative, one for autoimmune) summarizing all the publications supporting the association between them and herpesviridae. Those tables should at least include the references to the study, the summary of their key findings, the design of the study and the type of diseases studied. 

5) The author needs one or two schematic illustrations describing the pathophysiology of the diseases and in which part of the process, herpes may involve.

6) Figures 1 and 2 are not informative, please change them

7) Lines 94-192: Using bullet points and numbering is not appropriate for a review article. It can be used only for a technical document or lab procedures. Instead, please use those tables this reviewer suggested above. 

8) Lines 207-215: Please add this as a reference to the statement (PMID: 37374237)

9) I would like to suggest to invite one or more experienced scientists to be on board. I am sure their presence will bring more insights into the topic.

Comments on the Quality of English Language

No comment

Author Response

Comments and Suggestions for Authors

In this "review" article, Dr. De Francesco disscussed the possible association between herpes virus with neurodegenerative and autoimmune diseases. I do have some questions and concerns:

  • The author did not explain why neurodegenerative and autoimmune were picked? Are they interconnected? Please explain in the text and abstract.

An explanation has been provided

  • Similarly, why among many autoimmune diseases, SLE, MS and Sjogren were selected?

The choice derived by the involvement of some herpesviruses principally in these autoimmune diseases

  • Please rephrase the subheadings for example "How might herpesviruses contribute to the pathogenesis of these neurodegenerative diseases?" to make it more suitable for a conventional review article. Otherwise, this should be a perspective article, not a review.

The subheadings have been changed according to the reviewer’s suggestion

  • Please add tables (one for neurodegenerative, one for autoimmune) summarizing all the publications supporting the association between them and herpesviridae. Those tables should at least include the references to the study, the summary of their key findings, the design of the study and the type of diseases studied. 

The tables have been added according to the reviewer’s suggestion

  • The author needs one or two schematic illustrations describing the pathophysiology of the diseases and in which part of the process, herpes may involve.

The Figures have been added

  • Figures 1 and 2 are not informative, please change them

Figures 1 and 2 have been deleted

  • Lines 94-192: Using bullet points and numbering is not appropriate for a review article. It can be used only for a technical document or lab procedures. Instead, please use those tables this reviewer suggested above. 

Bullet points and numbering have been deleted according to the reviewer’s suggestion

  • Lines 207-215: Please add this as a reference to the statement (PMID: 37374237)

The reference has been added

  • I would like to suggest to invite one or more experienced scientists to be on board. I am sure their presence will bring more insights into the topic.

Thank you for your suggestion. I am conscious to be not an expert on this topic, but I was fascinated by the idea of analyzing how different herpesviruses could be involved in so many diseases and that was the reason why I tried to perform this review. I hope however that with your valuable suggestions it might be considered positively and improved in its content.

Round 2

Reviewer 1 Report

Comments and Suggestions for Authors

This revised review regarding the roles of  different herpesviruses in varous neurologic and autoimmune diseases is improved in comparison to the first review.  In particular instead of listing a long list of previous publications it has attempted to put the findings of these papers into paragraph form and has added some Tables to summarize the various findings.

The paper still needs extensive editing; the format of the paragraphs is not correct and many spaces are left between sentences. Also there are still a number of typos.  

Comments on the Quality of English Language

The authors need to get a native English speaker to edit the paper

Some of the words used do not appear to be the correct meaning intended by the authors

There are still numerous typos

The spacing between sentences and words is often incorrect

Reviewer 2 Report

Comments and Suggestions for Authors

First, I would like to convey my appreciation for the revised manuscript. Overall, it is improved in a way, but still much more to do to make it a readable manuscript. 

The authors need to invite one or more senior authors to help with the writing of this manuscript. It is not yet well arranged and the storyline is not easy to digest. 

In those previously requested tables, I can see some review articles included, and they are not acceptable as source of evidences. The authors need to only add original research to these tables. Additionally, the authors need to correctly specify the study design of those research. “Serological study” is not a study design. This is an example why a senior researcher is needed to be on board.

Comments on the Quality of English Language

No comment